# Hydrophobic Modification of Wood Using Tetramethylcyclotetrasiloxane

**DOI:** 10.3390/polym14102077

**Published:** 2022-05-19

**Authors:** Mingwei Tang, Xueren Fang, Bowen Li, Meng Xu, Haiyan Wang, Shuang Cai

**Affiliations:** Department of Chemical Engineering and Food Science, Hubei University of Arts and Science, 296 Longzhong Road, Xiangyang 441053, China; tangmingwei2022@163.com (M.T.); yqz520417@163.com (X.F.); bowenli2021@163.com (B.L.); 15039788785@139.com (M.X.)

**Keywords:** wood, hydrophobicity, surface modification, tetramethylcyclotetrasiloxane (D4H), anti-fouling

## Abstract

Hydrophobic surfaces have aroused considerable attention because of their extensive potential applications. In this work, we developed a facile strategy for fabricating hydrophobic and anti-fouling surfaces on wood substrates. The modification was accomplished simply by immerging wood into the tetramethylcyclotetrasiloxane (D4H) modifier solution for 5 min. The D4H modified wood was characterized using Fourier transform infrared spectroscopy, X-ray photoelectron spectroscopy, scanning electron microscope, and energy dispersive spectrometer. The result shows that the D4H modified wood had good hydrophobicity, and the water contact angle of wood in the radial and cross sections reached 140.1° and 152°. In addition, the obtained hydrophobic wood surface also showed anti-fouling properties, UV resistance and could withstand the tape peel test and finger wiping.

## 1. Introduction

Wood is the only renewable resource among the four internationally recognized raw materials. Due to its unique material properties and excellent environmental properties, wood is deeply loved by people [1,2,3,4]. However, wood has a rich pore structure and contains many hydrophilic hydroxyl groups [5,6,7]. Wood easily absorbs moisture in a humid environment. This can cause the wood to deform, discolor, rot and crack, thereby shortening the service life of the wood [8,9,10]. Therefore, in order to prolong the service life of wood, it is necessary to protect the wood.

In the past, chemical treatment of wood is a common method for enhancing water repellency and durability of wood. The chemical treatment of wood includes acetylation treatment [11,12,13], graft modification treatment [14,15,16,17,18], etc. These chemical treatments can effectively slow down the wood’s moisture absorption and extend its service. But, these modification methods cannot completely prevent moisture intrusion into the wood. Recently, constructing superhydrophobic surfaces on wood substrates is considered a promising approach to enhancing water repellency [19,20,21,22]. Numerous studies reported the successful construction of superhydrophobic coatings on wood substrates. Lin et al. obtained superhydrophobic wood using fumed silica and polymethylhydrosilane (PMHS) via a dehydrogenation reaction. The obtained modified wood has good superhydrophobic properties [5]. Yue et al. successfully deposited a superhydrophobic RTVSR coating onto the wood surface with SiO_2_ nanoparticles. The resulting superhydrophobic wood has good durability [23]. Jia et al., obtained remarkably robust superhydrophobic wood by using SiO_2_, commercial EP, and F13-TMS as a hydrophobic modifier. The resulting wood remains superhydrophobic after being subjected to long-term abrasion testing [24]. Yang et al., prepared superhydrophobic wood samples using polydopamine (PDA), 3-mercaptopropyltriethoxysilane (KH580) and nano-Al_2_O_3_ as the micro-nano structure maker. The obtained superhydrophobic wood showed excellent resistance to flow scouring, acid and base corrosion, and organic solvents [25]. However, these superhydrophobic wood surfaces have been prepared by complicated processing procedures and using expensive modifiers, which highly limit their industrial production. High hydrophobicity is sufficient to extend the service life of wood, and it does not need to be superhydrophobic.

Here, highly hydrophobic wood has been prepared by a simple, efficient, one-step method. Briefly, tetramethylcyclotetrasiloxane (D4H) was chosen from industrial raw materials to graft hydrophobic –CH_3_ groups on wood substrates. D4H is a reactive siloxane containing silicon-hydrogen (–Si–H) bonds that can undergo addition reactions with unsaturated olefins. Mainly used for the preparation of various polymethyl hydrogen siloxanes with specific hydrogen content and functional group-modified polysiloxanes. In this work, the modification process is very simple: by dipping the wood in D4H modification solution it can be obtained highly hydrophobic wood with a water contact angle (WCA) of 140.1° and withstand UV radiation and anti-fouling properties.

## 2. Materials and Methods

### 2.1. Materials

Tetramethylcyclotetrasiloxane (D4H) was obtained from Xiamen Guiyou New Material Technology Co., Ltd. (Fujian, China). Kastredt catalyst was provided by Dongguan Maiteng Rubber and Plastic Materials Co., Ltd. (Guangdong, China). N-hexane was obtained from Sinopharm Chemical Reagent Co., Ltd. (Shanghai, China). For this study, the Chinese fir (sapwood) with a density of 340 ± 40 kg/m^3^) was purchased from the Guangdong province of China. The dimensions of the wood sample are 20 mm × 20 mm × 10 mm. The moisture content of the wood was 12%.

### 2.2. Hydrophobic Modification of Wood Substrates

Modified fluid with D4H concentrations of 0%, 1%, 5%, 10%, 15%, 20%, and 25% were prepared by mixing the D4H, n-hexane, and catalyst together. The formula for D4H modifier solutions is shown in Table 1. The wood samples were immerged into the D4H solutions for 5 min and then withdrawn. Then, the D4H-treated wood specimens were finally washed with n-hexane to remove the D4H not covalently bonded onto the wood surface. Finally, the resulting wood sample was dried at 100 °C for 30 min.

### 2.3. Characterization

The surface morphology and microstructure of the samples were performed by scanning electron microscopy (SEM, S-4800, Hitachi, Japan). The elemental analysis of the wood surface was performed through an energy dispersive spectrometer (EDS, EX-250, Horiba, Japan). Infrared spectrometer (IRPrestige-21, Shimadzu, Japan) and X-ray photoelectron spectroscopy (XPS, Thermo Scientific Escalab 250Xi, Waltham, MA, USA) were tested to characterize the functional group composition and changes in the surface chemistry of the D4H modified wood. The woods’ water contact angle (WCA) was measured using a 5 μL distilled water droplet on a contact angle meter (Powereach JC2000D1; Shanghai, China). After the water droplet stayed on the wood surface for 30 s, the WCA was measured. The WCA was obtained as averages of measurements taken from at least five different positions on each sample’s surface. In addition, the 24-h water absorption test and abrasion resistance tests (finger wiping, and tape peeling tests) of the D4H modified wood samples were characterized according to the previous research reports [17,18,26,27].

## 3. Results and Discussion

### 3.1. The Mechanism of D4H Modified Wood

The mechanism of D4H modification to prepare hydrophobic wood is: the hydrophilic –OH groups on the wood surface will undergo a dehydrogenation reaction with the –Si–H bonds on the D4H structure in the presence of catalysts, so that the D4H with hydrophobic methyl (–CH_3_) groups will be grafted onto the wood surface to finally obtain hydrophobized wood. As shown in Figure 1a, D4H is a reactive siloxane containing four Si–H participates in various chemical reactions, especially with unsaturated olefins and –OH groups containing materials, such as wood with abundant hydroxyl groups. In addition, the methyl group in the D4H structure makes it have low surface energy. As shown in Figure 1b, the surface of unmodified wood is composed of a large number of hydrophilic –OH groups. In the presence of a Kastredt catalyst, the –OH groups on the wood surface undergo an ultrafast dehydrogenation reaction with D4H. As shown in Appendix A (see Appendix A), after immersing the wood samples in the D4H modifier solution, a large number of air bubbles was seen. As shown in Figure 1c, through this simple modification method, it can be speculated that D4H is covalently grafted onto the wood surface. Consequently, the resulting wood samples will have good hydrophobic properties.

### 3.2. Change in Chemical Properties of Wood

FTIR spectrum of wood before and after D4H modification is shown in Figure 2. By comparison, the samples before and after D4H modification have some same bands. Bands at 3411 cm^−1^ are attributed to the stretching vibration of the –OH group [1]. The absorption peak at 2916 cm^−1^ belongs to the stretching vibration of the –CH_3_ groups [18]. And the absorption peaks at 1743 cm^−1^ and 1465 cm^−1^ are assigned to the C=O stretching vibration and C=C asymmetric stretching vibration [2], respectively. In addition to the same absorption peaks, D4H-modified wood samples appeared with new absorption peaks. The absorption peak at 2168 cm^−1^ can be ascribed to the –Si–H of the D4H structure [28,29]. The band at 900 cm^−1^ is attributed to the –Si–CH_3_ absorption peaks [18]. The absorption peak at 762 cm^−1^ is assigned to the Si–O–Si bending mode from the D4H structure [1,30]. Combined with the experimental phenomenon of Appendix A (see Appendix A), it can be further speculated that D4H was grafted on the wood surface in the form of chemical bonds.

To further determine the surface chemistry of wood before and after D4H modification, an XPS test was carried out. As shown in Figure 3a,b, both unmodified and D4H-modified wood (25% D4H content) samples have C1s (285.17 eV) and O1s (531.18 eV) characteristic signal peaks [31]. In addition, D4H-modified wood samples appeared with characteristic signal peaks of Si, which were located in 103.28 eV (Si2p) and 154.28 eV (Si2s), respectively. This may be attributed to the grafting of D4H on the wood surface. To better research the surface functional group composition of D4H-modified wood and untreated wood, the high-resolution O1s spectra were resolved into spectra of various oxygen components. As shown in Figure 3c,d, compared to untreated wood, the D4H treated wood showed a new O2 (Si–O–Si or Si–O–C bonds) signal, which appeared at about 532.46 eV. The appearance of the O2 signal can further speculate that D4H was grafted on the wood surface in the form of chemical bonds [18].

### 3.3. Morphological Observation and Elemental Composition Analysis

Changes in surface morphology and chemical composition of wood before and after D4H treatment were determined by the SEM-EDS test. As shown in Figure 4a, the surface of unmodified wood is smooth and rich in micron-sized pores. After surface modification with D4H (25% content), we can see that there is no obvious change in the surface morphology of the modified wood, which indicates that the D4H modification has no effect on the microstructure. The EDS results are presented as an inset in Figure 4a,b. The EDS analysis of the original wood reveals that it contains C (61.93%) and O (38.07%) atoms, and the Si element was not detected. In contrast, D4H wood samples exhibit the presence of the Si element. In a word, these results (FTIR, XPS and SEM-EDS) can confirm that D4H is covalently grafted on the wood surface.

### 3.4. Hydrophobicity of Wood

As shown in Figure 5, with the increase in D4H concentration, the WCA of modified wood also increased gradually. The temperature and humidity of the environment during the contact angle test were 25 °C and 60%, respectively. The WCA of the untreated wood at radial and cross-sections are only 27.8° and 0°, respectively, indicating that the unmodified wood is very hydrophilic. As the D4H concentration increased from 0% to 1%, the WCA of the treated wood in radial and cross-sections increased to 130.9° and 149.5°, respectively. This is due to the grafting of hydrophobic –CH_3_ groups (from D4H) on the surface of the modified wood. According to Figure 5, when the D4H concentration was increased from 10% to 25%, the WCA of the modified wood in the radial and cross-sections did not change significantly. This can be attributed to the fact that the wood surface has been completely covered by D4H with low surface energy, there is no site to continue grafting D4H, so the hydrophobicity will not change significantly. It should be noted here that D4H modification of wood surfaces is very efficient. This is mainly attributed to the high reactivity of –OH groups on the wood surface with D4H in the presence of catalysts. As shown in Appendix A (see Appendix A), when the unmodified wood sample was immersed in the D4H modifier solution, a large number of bubbles can be seen, which is due to the dehydrogenation reaction of the wood and D4H.

### 3.5. The Durability Test of D4H Modified Wood Surface

To demonstrate the practicality of D4H-modified wood, abrasion resistance tests were carried out. As shown in Figure 6a, the abrasion resistance performance of D4H-modified (25% content) wood samples were evaluated by a finger wiping test. Press a finger firmly on the surface of the D4H-modified wood, and then drop water droplets on the surface to observe the shape of the water droplets. Clearly, the D4H-modified wood samples still have good hydrophobic properties after wiping them with a finger. The water droplets dyed with methyl blue are spherical on the wiped surface of the D4H-modified wood, with a WCA of 139.4°. As shown in Figure 6b, abrasion resistance of D4H modified wood was further evaluated by a tape peel test. The tape was glued to the D4H modified (25% content) wood surface and tightly pressed several times, and then the tape was taken off. And then test the WCA of the wood surface after the tape is peeled off. The results show that the D4H-modified wood still has good hydrophobic properties. In addition, we also evaluated the chemical stability of D4H-modified wood (shown in Appendix A).

To further evaluate the durability of the hydrophobic layer on the surface of D4H-modified wood, a UV resistance test was carried out. As shown in Figure 7, the D4H modified (25% content) wood samples were placed under a UV lamp with a power of 36 W (wavelength: 356 nm) and irradiated for different times (0–72 h). As can be seen from the figure, after 72 h of UV irradiation, the D4H modified wood samples still have good water repellency, and the WCA at the cross-section and radial section was still greater than 149° and 136°. It means that the D4H-modified wood possessed good UV durability.

### 3.6. Water Absorption

Figure 8 shows the water absorption test chart of unmodified wood and D4H-modified wood after soaking in water for 24 h. Obviously, the 24-h water absorption value of wood samples modified with D4H is much lower than that of unmodified wood (109.6%). When the concentration of the D4H modifier exceeds 15%, the modified wood has a lower water absorption value. And, when the D4H concentration reaches 25%, the water absorption value of modified wood reaches the lowest value (32.6%). The result shows that D4H modification can impart wood with good water repellency.

### 3.7. Anti-Fouling Performance Test

In this study, the D4H-modified wood did not achieve superhydrophobicity in the radial section. The radial section of wood is widely used in daily life. This means that the wood obtained by D4H treatment does not have self-cleaning properties. Nevertheless, a certain level of anti-fouling effect is enough for some wood products used indoors. After D4H modification, the surface of the resulting wood sample was grafted with hydrophobic –CH_3_ groups, which reduced the surface energy of the wood and made the wood have good anti-fouling performance. Figure 9 shows the anti-fouling resistance test images of unmodified wood and D4H modified. Put soy sauce, milk, and orange juice droplets on the surface of unmodified and modified wood, then use a paper towel to absorb the contamination to see if it will leave marks on the surface. Obviously, the unmodified wood is wetted by milk, soy sauce, and orange juice indicating that the unmodified wood is heavily polluted. As a comparison, the wood samples obtained by D4H modification have no residual contaminants on the surface. Therefore, it can be concluded that D4H-modified wood had good anti-fouling properties to common household pollutants.

## 4. Conclusions

In summary, highly hydrophobic wood with good anti-fouling properties was successfully fabricated by simple soaking in a D4H modifier solution. The proposed formation mechanism of hydrophobicity on the wood surface resulted from D4H with low surface energy covalently grafting on the surface of the wood. The WCA of the D4H treated wood was increased significantly compared to that of untreated wood, and the water absorption of the D4H-treated (25% content) wood samples was lower than 40% after 24 h of water immersion. In addition, the obtained hydrophobic wood showed good UV resistance and wear resistance, which indicates promising application prospects in numerous fields.

## Figures and Tables

**Figure 1 polymers-14-02077-f001:**
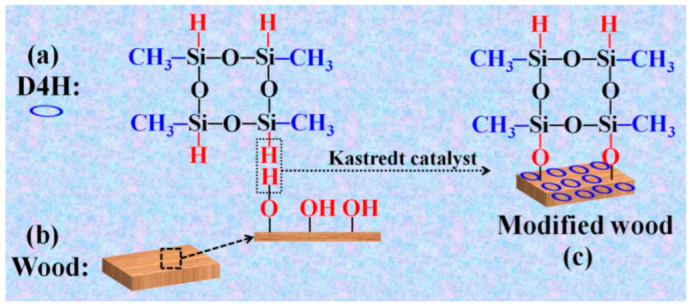
Schematic representation of the process for obtaining the D4H modified wood: (**a**) D4H structure; (**b**) Wood; (**c**) Modified wood.

**Figure 2 polymers-14-02077-f002:**
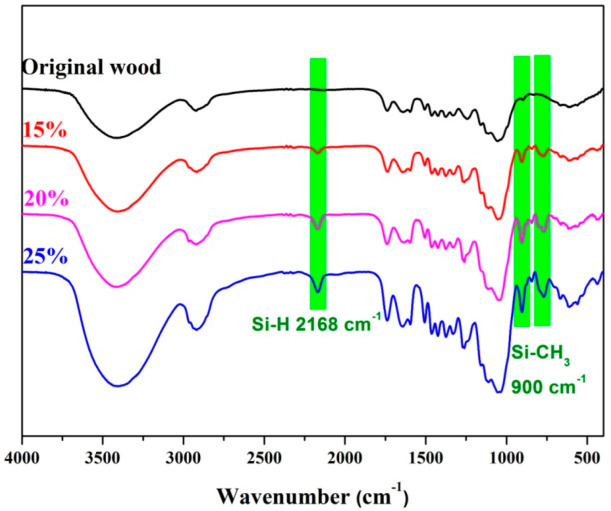
FTIR spectra of wood before and after D4H modification.

**Figure 3 polymers-14-02077-f003:**
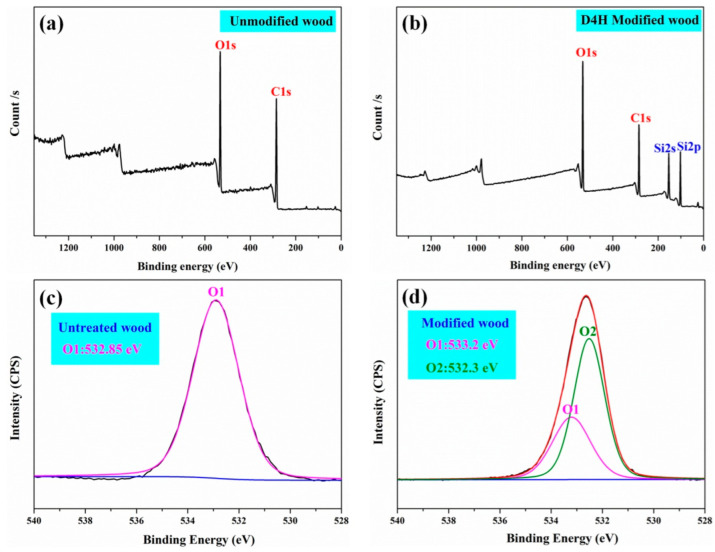
XPS spectra of untreated (**a**) and 25% D4H treated wood (**b**) and high-resolution spectra of O1s (**c**) and (**d**).

**Figure 4 polymers-14-02077-f004:**
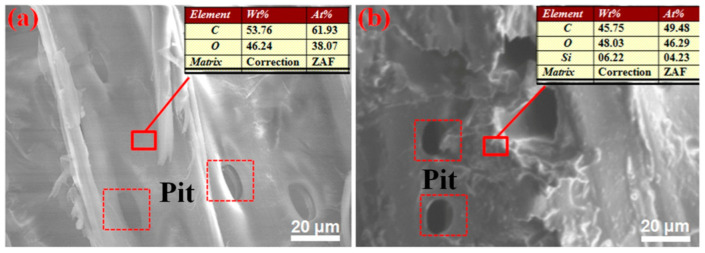
SEM images and EDS analysis of wood before (**a**) and after (**b**) 25% D4H modification.

**Figure 5 polymers-14-02077-f005:**
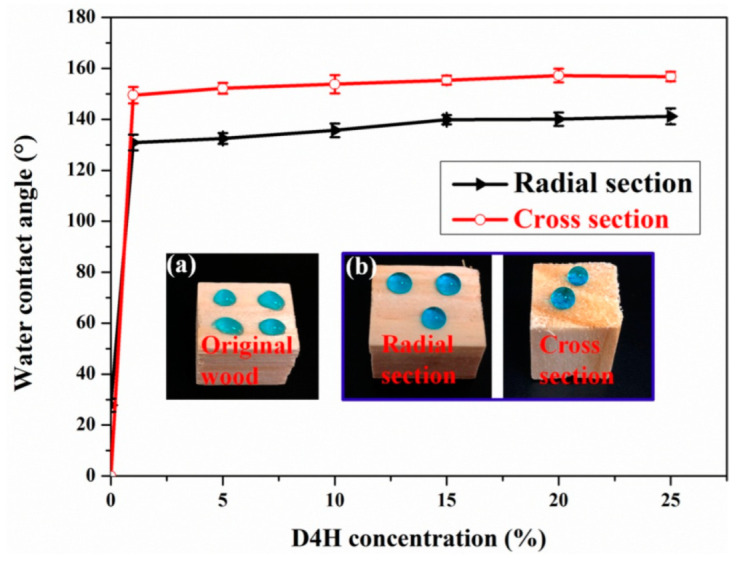
WCA on the surface of wood samples treated with different D4H concentrations. Photographs of water droplets (dyed with methylene blue) on the original (radial section) (**a**) and D4H modified wood (cross section and radial section) (**b**).

**Figure 6 polymers-14-02077-f006:**
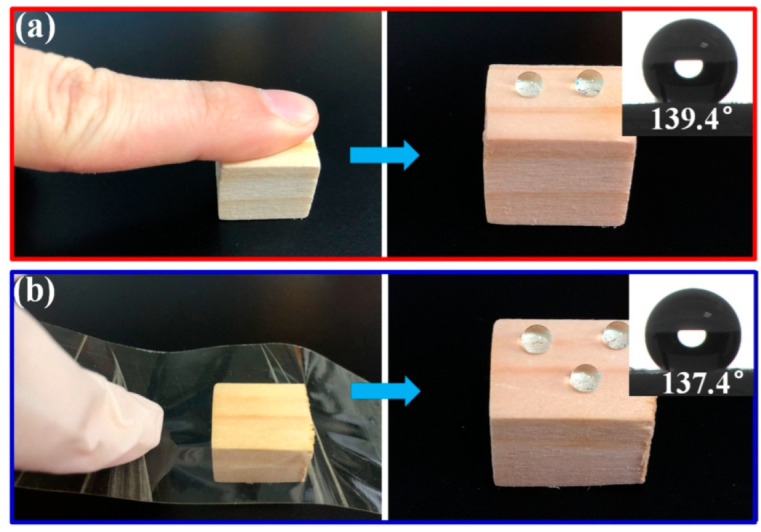
Images of finger wiping test (**a**) and tape peeling (**b**).

**Figure 7 polymers-14-02077-f007:**
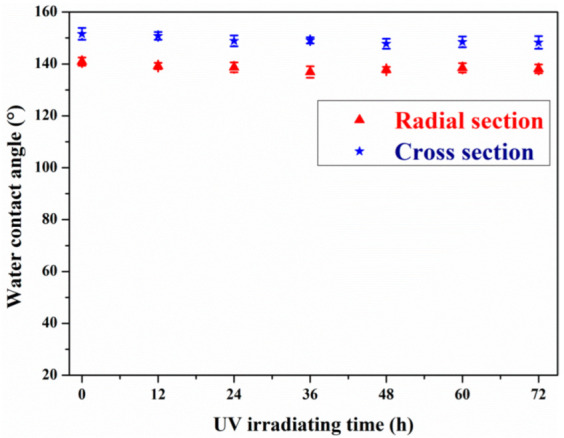
UV radiation resistance test of D4H modified wood.

**Figure 8 polymers-14-02077-f008:**
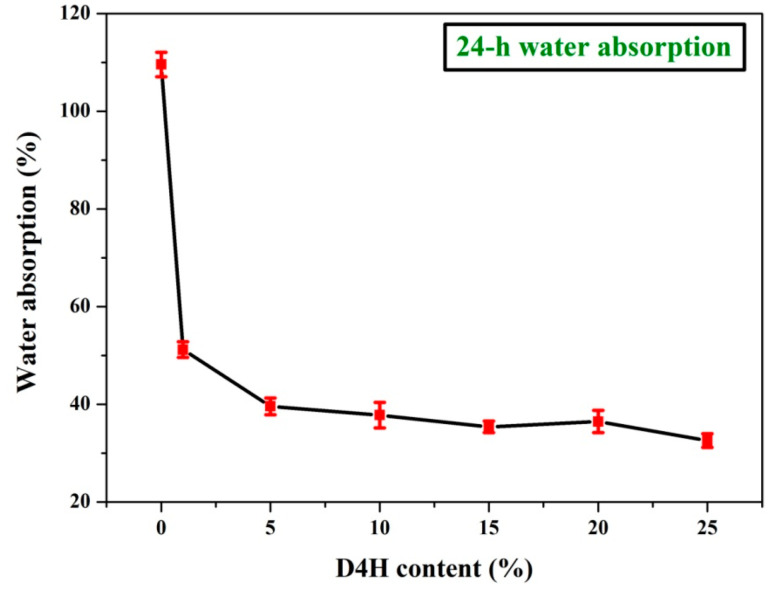
24-h water absorption test before and after wood modification.

**Figure 9 polymers-14-02077-f009:**
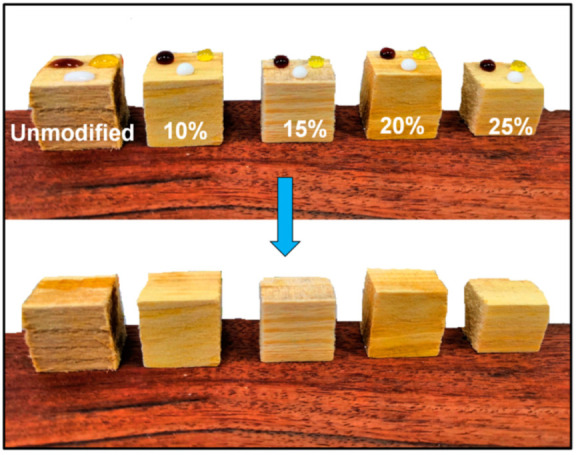
Anti-fouling property of untreated and D4H (different contents) treated woods to milk, soy sauce, and orange juice.

**Table 1 polymers-14-02077-t001:** Formula for D4H modifier solutions.

D4H Contents	D4H /g	n-hexane/g	Kastredt Catalyst/mL
0%	0.0	50.0	0.15
1%	0.5	49.5	0.15
5%	2.5	47.5	0.15
10%	5	45	0.15
15%	7.5	42.5	0.15
20%	10	40.0	0.15
25%	12.5	37.5	0.15

## Data Availability

The data presented in this study are available on request from the corresponding author.

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
