# Peer review of "Hydrophobic Modification of Wood Using Tetramethylcyclotetrasiloxane"

_polymers, 2022, doi:10.3390/polym14102077_

Round 1

Reviewer 1 Report

Hydrophobic modification of wood using tetramethylcyclo-tetrasiloxane

Modification of the wood surface in order to make it hydrophobic is one of the important directions of wood research. These issues have been the subject of many studies. Nevertheless, the research conducted as part of the reviewed paper broadens the scope of knowledge in this topic. In this context, the subject of the article should be considered interesting and up-to-date. Nevertheless, the authors should clarify a few points and make some adjustments:

  • There is no information in the literature review on what tetramethylcyclotetrasiloxane is currently used for and whether it has already been used for wood modification.
  • No characteristics of the modified wood material - wood density, humidity, sample size, share of sapwood and heartwood, etc. The authors only state that they used Chinese fir.
  • The description of the modification process should be supplemented and clarified. The authors mention the addition of the catalyst, but do not specify its share in the modifying mixture.
  • There is no information about the effectiveness of the modification, e.g. the percentage increase in the mass of the samples.
  • The description of individual research methods should be supplemented. No information on the number of repetitions. In the case of determining the contact angle, there is no information about how long after the drop was placed, the determination was made. There is no information on the methodology of abrasion resistance tests or tape peeling - the authors cite only literature sources. There is no information on the conditions for the implementation of the UV resistance test - the methodology is partially described only in the discussion of the results. There is no information on the conditions for the implementation of the anti-fouling performance test.
  • No information is available, why in Figure 2 there are no results for variants 1%, 5%, 10%.
  • There is no information as to which modification variants were tested with the use of X-ray photoelectron spectroscopy (XPS). In Figure 3, only general information about wood modification is given, without specifying the variant.
  • In Figure 4 and in the description, there is no information which modification variant the presented results (photos) relate to.
  • Figure 5 shows that unmodified wood has a contact angle of 0 degrees. This is not in line with the photo marked "a" shown in this figure.
  • With regard to the results of the finger wiping test or tape peeling presented in figure 6, there is no information about which variants of the modifications apply to.
  • With regard to the results of the UV radiation resistance test presented in Figure 7, there is no information about which modification variants apply to.
  • In figure 8, there are no standard deviations for the presented results of water absorption.
  • The results presented in Figure 9 do not give a clear answer as to the effects of the Anti-fouling performance test.
  • The conclusions are very general. In addition, the authors state, for example, that the content of D4H in wood increased to 25%, although they did not confirm it. The authors only used a solution with a concentration of 25% for modification, which is not synonymous with an increase in the content to 25%.

Taking into account the above remarks and suggestions, the article requires a major revision.

Author Response

Response to Reviewer 1 Comments

Point 1: There is no information in the literature review on what tetramethylcyclotetrasiloxane is currently used for and whether it has already been used for wood modification.

Response 1: Thank you very much for your suggestion. We have added it in the new submitted manuscript. In addition, by consulting the literature, we would like to explain that tetramethylcyclotetrasiloxane (D4H) has not been used in the field of wood hydrophobic modification.

Point 2: No characteristics of the modified wood material - wood density, humidity, sample size, share of sapwood and heartwood, etc. The authors only state that they used Chinese fir.

Response 2: Thank you very much for your suggestion. We have added it in the new submitted manuscript.

Point 3: The description of the modification process should be supplemented and clarified. The authors mention the addition of the catalyst, but do not specify its share in the modifying mixture.

Response 3: Thank you very much for your suggestion. We have corrected it in the new submitted manuscript.

Point 4: There is no information about the effectiveness of the modification, e.g. the percentage increase in the mass of the samples.

Response 4: Thank you very much for your suggestion. In fact, this study mainly proposes a method for hydrophobic modification of wood, by dipping the wood in D4H modification solution it can be obtained highly hydrophobic wood. It imparts water repellency to wood through simple surface chemical modification. Therefore, the mass of wood before and after modification is basically unchanged.

Point 5: The description of individual research methods should be supplemented. No information on the number of repetitions. In the case of determining the contact angle, there is no information about how long after the drop was placed, the determination was made. There is no information on the methodology of abrasion resistance tests or tape peeling - the authors cite only literature sources. There is no information on the conditions for the implementation of the UV resistance test - the methodology is partially described only in the discussion of the results. There is no information on the conditions for the implementation of the anti-fouling performance test.

Response 5: We deeply appreciate the reviewer's suggestion in such detail. We have added these details in the new submitted manuscript. In fact, in the field of wood hydrophobic modification, there is no standard test method for evaluating the durability of hydrophobic coatings on wood surfaces. In this study, we briefly evaluated the durability (24-h water absorption test, finger wiping test and tape peeling tests) of hydrophobic coatings on D4H modified wood surfaces based on the research reports of other scholars (Wu Y , Jia S , Shuang W , et al. A facile and novel emulsion for efficient and convenient fabrication of durable superhydrophobic materials. Chemical Engineering Journal, 2017, 328) and (Lin, W.; Huang, Y.; Li, J.; Liu, Z.; Yang, W.; Li, R.; Chen, H.; Zhang, X. Preparation of highly hydrophobic and anti-fouling wood using poly(methylhydrogen)siloxane. Cellulose 2018, 25, 7341-7353).

Point 6: No information is available, why in Figure 2 there are no results for variants 1%, 5%, 10%.

Response 6: Thank you very much for your suggestion. In fact, when the D4H concentration is low, the grafted D4H on the wood surface will also be relatively small, and the characteristic absorption peak of the –Si-H bond on the surface of D4H-modified wood does not appear obviously during the FTIR test. Therefore, only the infrared spectra of 15%, 20%, and 25% D4H modified wood are provided.

Point 7: There is no information as to which modification variants were tested with the use of X-ray photoelectron spectroscopy (XPS). In Figure 3, only general information about wood modification is given, without specifying the variant.

Response 7: We deeply appreciate the reviewer's suggestion and comment. We have added it in the new submitted manuscript.

Point 8: In Figure 4 and in the description, there is no information which modification variant the presented results (photos) relate to.

Response 8: Sorry for our mistake. We have added it in the new submitted manuscript.

Point 9: Figure 5 shows that unmodified wood has a contact angle of 0 degrees. This is not in line with the photo marked "a" shown in this figure.

Response 9: Sorry for our mistake. The photograph of Figure 5(a) is an image of a radial section of untreated wood. So it has a certain water contact angle. The WCA of the untreated wood at radial and cross-sections are only 27.8° and 0°, respectively. The cross-section of the unmodified wood is rough and has strong hygroscopicity. The contact angle measured by the experiment is 0 degrees.

Point 10: With regard to the results of the finger wiping test or tape peeling presented in figure 6, there is no information about which variants of the modifications apply to.

Response 10: Sorry for our mistake. We have added it in the new submitted manuscript.

Point 11: With regard to the results of the UV radiation resistance test presented in Figure 7, there is no information about which modification variants apply to.

Response 11: Sorry for our mistake. We have added it in the new submitted manuscript.

Point 12: In figure 8, there are no standard deviations for the presented results of water absorption.

Response 12: Thank you very much for your suggestion. We have corrected it in the new submitted manuscript.

Point 13: The results presented in Figure 9 do not give a clear answer as to the effects of the Anti-fouling performance test.

Response 13: Thank you very much for your suggestion. We have corrected it in the new submitted manuscript.

Point 14: The conclusions are very general. In addition, the authors state, for example, that the content of D4H in wood increased to 25%, although they did not confirm it. The authors only used a solution with a concentration of 25% for modification, which is not synonymous with an increase in the content to 25%.

Response 14: Thank you very much for your suggestion. We have corrected it in the new submitted manuscript.

Reviewer 2 Report

Comments:

The paper prepared hydrophobic wood by a simple, efficient, one-step method using tetramethylcyclotetrasiloxane. There are several questions that need to be revised. The detailed comments are as follows:

  • Introduction: All the reference years are in 2020 and before, and there is only a few are in 2021. The author should cite the reference in recent years to illustrate the research progress at this stage. Yang, H. Li, Z. Yi, M. Liao, Z. Qin, Colloids and Surfaces A: Physicochemical and Engineering Aspects, 637 (2022) 128219.This article is obviously more suitable for your article and more novel than reference [24].
  • Materials and methods:The detailed preparation method of modified fluid is not clearly described in hydrophobic modification of wood substrates.
  • Change in chemical properties of wood: The FTIR spectrum at 1100cm-1, 1600cm-1, 2700cm-1 of wood before and after D4H modified are not explained clearly. Does the modification affect the performance of wood itself.
  • Hydrophobicity of wood: As the D4H concentration increased from 0% to 1%, wood has begun to become hydrophobic. Why added to 15% and how to explain the relationship between the addition amount and hydrophobic property.
  • Hydrophobicity of wood: The temperature and humidity of contact angle test should be indicated and stability research should be added to ensure that the layer will not fall off. For superhydrophobic wood, the sliding angle (SA) is also one of the indicators of wood hydrophobicity, which should be supplemented.
  • The durability test of D4H modified wood surface: Abrasion resistance tests carried in paper is not enough to prove its wear resistance. It is necessary to grind several times with a sander.
  • Water absorption: Water repellency and UV durability are not necessary for hydrophobic wood and the effect is not good enough.
  • Anti-fouling performance test: Fig 9. seems not indicating that D4H modification can impart good anti-fouling property to wood.

Author Response

Response to Reviewer 2 Comments

Point 1: Introduction: All the reference years are in 2020 and before, and there is only a few are in 2021. The author should cite the reference in recent years to illustrate the research progress at this stage. Yang, H. Li, Z. Yi, M. Liao, Z. Qin, Colloids and Surfaces A: Physicochemical and Engineering Aspects, 637 (2022) 128219.This article is obviously more suitable for your article and more novel than reference [24].

Response 1: We deeply appreciate the reviewer's suggestion in such detail. We have corrected it in the new submitted manuscript.

Point 2: Materials and methods: the detailed preparation method of modified fluid is not clearly described in hydrophobic modification of wood substrates.

Response 2: Thank you very much for your suggestion. We have corrected it in the new submitted manuscript.

Point 3: Change in chemical properties of wood: The FTIR spectrum at 1100cm-1, 1600cm-1, 2700cm-1 of wood before and after D4H modified are not explained clearly. Does the modification affect the performance of wood itself.

Response 3: Thank you very much for your suggestion. In fact, through the test and analysis of the FTIR, we are to confirm that new absorption peaks will appear after the surface chemical modification of wood by D4H. Moreover, D4H modification treatment will not change the original functional groups of wood.

Point 4: As the D4H concentration increased from 0% to 1%, wood has begun to become hydrophobic. Why added to 15% and how to explain the relationship between the addition amount and hydrophobic property.

Response 4: Thank you very much for your suggestion. In fact, After modification, there are D4H structure with very low surface energy grafted onto the wood surface, which increased the wood WCA significantly. More importantly, it should be pointed out that the D4H modification of the wood surface is highly efficient. A good hydrophobicity could be realized even though the content of D4H in the modifier solution is only 1%. This is attributed to the highly reactive activity between the D4H and hydroxyl groups on the wood surface with Kastredt catalyst. At this time, a large amount of D4H has been grafted on the surface of the wood, so the wood has good hydrophobicity. If the concentration of D4H continues to increase, the hydrophobicity of the modified wood will not change significantly. This can be attributed to the fact that the wood surface has been completely covered by D4H with low surface energy, there is no site to continue grafting D4H, so the hydrophobicity will not change significantly.

Point 5: Hydrophobicity of wood: The temperature and humidity of contact angle test should be indicated and stability research should be added to ensure that the layer will not fall off. For superhydrophobic wood, the sliding angle (SA) is also one of the indicators of wood hydrophobicity, which should be supplemented.

Response 5: We deeply appreciate the reviewer's suggestion in such detail. We have corrected it in the new submitted manuscript. In addition, we would like to point out that the modified wood prepared in this study did not achieve superhydrophobicity at the radial section. The longitudinal surfaces of wood are most relevant in practical applications.  Therefore, we did not test the sliding angle with modified wood.

Point 6: The durability test of D4H modified wood surface: Abrasion resistance tests carried in paper is not enough to prove its wear resistance. It is necessary to grind several times with a sander.

Response 6: Thank you very much for your suggestion. I agree with your opinion. However, this study mainly proposes a method for hydrophobic modification of wood, by dipping the wood in D4H modification solution it can be obtained highly hydrophobic wood. Afterwards, we performed a simple evaluation of the durability of the hydrophobic layer based on the literature. Moreover, D4H is mainly grafted on the wood surface in the form of covalent bonds, forming a thin hydrophobic layer. Therefore, we did not conduct a very detailed study on the durability of the hydrophobic film on the wood surface, but only briefly evaluated the durability of the hydrophobic film on the D4H modified wood surface based on the research reports of others.

Point 7: Water absorption: Water repellency and UV durability are not necessary for hydrophobic wood and the effect is not good enough.

Response 7: Thank you very much for your suggestion. In fact, the 24-hour water absorption value of wood samples (32.6%) modified with D4H is much lower than that of unmodified wood (109.6%). I think this result shows that D4H modification can significantly improve the water repellency of wood. In addition, the UV resistance test is to evaluate the resistance stability of the hydrophobic film layer on the modified wood surface. The results showed that the modified wood still had good hydrophobicity after 72-h UV irradiation.

Point 8: Anti-fouling performance test: Fig 9. seems not indicating that D4H modification can impart good anti-fouling property to wood.

Response 8: Thank you very much for your suggestion. In fact, Figure 9 showed the images of unmodified and modified woods with milk, soy sauce, and orange juice droplets on their surface and the images of them after absorbing the drops off with a tissue. For unmodified wood, all of radial section was well-wetted by milk, soy sauce, and orange juice. On the contrary, milk, soy sauce, and orange juice droplets displayed good spherical shapes. Most importantly, the surface of D4H modified wood is dry and not polluted at all. So, I think D4H modification can impart good anti-fouling property to wood.

Reviewer 3 Report

The paper Hydrophobic modification of wood using tetramethylcyclotetrasiloxane proposes a strategy for obtaining hydrophobic and anti-fouling surfaces on wood. The idea is to modify wood surface by D4H. A series of tests have been applied to verify the effectiveness of wood modification in terms of increased hydrophobicity and anti-fouling properties.

The paper is potentially interesting, but I think that it not suitable, in the present form, to be published in Polymers.

  • English language must be revised, several problems were found. For example:

- But, because of wood is a polymer material composed of cellulose, hemicellulose and lignin. (the sentence is incomplete)

- Consequence, in order to prolong the service … (Consequence?)

- … was chose from industrial …. (chosen)

- The modification process is very simple just simply dipping the wood in D4H modification solution can obtain highly hydrophobic wood. (The modification process is very simple: just dipping the wood in D4H solution allows for obtaining highly hydrophobic wood or The modification process is very simple: by dipping the wood in D4H modification solution it can be obtained highly hydrophobic wood.

- As a result, the resulting wood has good…. (remove this part). Simply continue the previous sentence: The modification process is very simple: by dipping the wood in D4H modification solution it can be obtained highly hydrophobic wood with a water contact angle (WCA) of 140.1° and withstand UV radiation and anti-fouling properties.

- Hydrophobic modification of wood surface can be achieved by simply immersing wood in D4H modification solution for 5 minutes. (modification … modification, wood … wood; please avoid repetitions by simply write in D4H solution and immersing samples)

- Observation of surface micromorphological changes before and after wood modification by scanning electron microscopy (SEM, S4800). (The verb is lacking in this sentence. For example you can use: was performed by scanning electron microscopy)

- The analysis of wood was characterised through X-ray spectroscopy (EDS, EX-250).

Analysis was made or was performed not was characterised. EDS is energy dispersive spectroscopy.

- Etc. Etc.

  • In the abstract generally is better to avoid abbreviation (SEM-EDS, XPS, FTIR).
  • This paragraph is very poor: the techniques are not explained, the measurement modalities are not detailed, the experimental conditions are not reported.
  • As shown in movie S1. There is not any movie annexed to the paper. Anyway I think that it is not useful for the comprehension of the research.
  • Caption of figure 1 should be: Schematic representation of the process for obtaining the D4H modified wood
  • And the absorption peaks at 1743 cm-1 and 1465 cm-1 were belongs (belong or can be attributed/assigned)! Later in the same paragraph: at 2168 cm-1 is belongs (belongs or is assigned)! At 762 cm-1 is belongs (belongs or is assigned)!

Please revise English language.

  • … O1s spectra was resolved into spectra of various … Spectra is plural so you have to use were resolved.
  • … showed a new O2 …. Appeared at about 532.46 eV. I suggest to use the present for the verbs because you are showing the present results.
  • In contrast, D4H treated wood samples appeared Si element.

It should be: In contrast, D4H wood samples exhibit the presence of Si element.

  • In the inset of figure 4 the chemical elements are reported with the letter K that, I suppose, is the line (Kα). I suggest to remove it or to write in the correct form (for example with a space between the letter of the chemical element and K).
  • The sentence: The WCA could represent the hydrophobicity, it quite obvious and I suggest to remove it.
  • Did not change much. It is better to write do not change significantly.
  • The durability test of D4H modified wood surface.

Firstly, and explanation of the test must be supplied in Materials and Methods.

Secondly, I have great doubts about the finger wiping test. It is a completely arbitrary evaluation that cannot be reproduced and repeated because it depends on uncontrolled factors.

According to my opinion it is not a scientific test and I don’t know (I made a research on this) standards reporting the finger test.

I propose to remove this test.

The tape peeling test, on the other hand, is a standard test that can be used to evaluate the resistance to peeling of a treated surface. Peel tests are used to assess bond quality since the predominant stress, as the name suggests, is peel or tension.

But the tape must be weighted before and after the test (the dimension of the tape must be precise) in order to have a scientific datum to affirm that no material has been removed from the surface.

Please, repeat this test by weighting the tape and check the international standards for the test.

  • Moreover, the colour change of D4H modified wood was significantly decreased compared to the original wood after UV irradiation.

Did the authors make colour measurements before and after ageing?

No information about this is reported in Materials and Methods: nor for the UV irradiation nor for the colour measurements.

The authors must add such experimental part in Materials and Methods, or they have to remove it from the results.

Moreover, the sentence is completely wrong. I suppose it should be: Moreover, the colour changes due to UV irradiation, in D4H modified wood are much lower that those undergone in unmodified wood.   

  • Water absorption

How this test has been performed? The authors must explain it in Materials and Methods.

This test seems to me quite unnecessary.

  • Anti-fouling test

Also in this case the authors use an empiric test.

How did the authors evaluate the absence/presence of contaminants on the surfaces of wood samples? Only by visual observation?

This seems to me no scientific and not acceptable in a paper published on Polymer.

So, or the authors use scientific methods for evaluated the presence or absence of residues, otherwise this test must be removed.

In conclusion, the paper is potentially interesting, but it must be deeply revised to have a possibility of being published in Polymers.

Author Response

Response to Reviewer 3 Comments

We deeply appreciate the reviewer's suggestion and comment. Based on your comments and suggestions, we have made a lot of revisions in the new submitted manuscript.

Point 1: The paper is potentially interesting, but I think that it not suitable, in the present form, to be published in Polymers.

Response 1: Thank you very much for your suggestion. we have made a lot of revisions in the new submitted manuscript.

Point 2: But, because of wood is a polymer material composed of cellulose, hemicellulose and lignin. (the sentence is incomplete).

Response 2: Thank you very much for your suggestion. We have corrected it in the new submitted manuscript.

Point 3: Consequence, in order to prolong the service … (Consequence?).

Response 3: Thank you very much for your suggestion. We have corrected it in the new submitted manuscript.

Point 4:… was chose from industrial …. (chosen).

Response 4: Thank you very much for your suggestion. We have corrected it in the new submitted manuscript.

Point 5: The modification process is very simple just simply dipping the wood in D4H modification solution can obtain highly hydrophobic wood. (The modification process is very simple: just dipping the wood in D4H solution allows for obtaining highly hydrophobic wood or The modification process is very simple: by dipping the wood in D4H modification solution it can be obtained highly hydrophobic wood.

Response 5: We deeply appreciate the reviewer's suggestion in such detail. We have corrected it in the new submitted manuscript.

Point 6: As a result, the resulting wood has good…. (remove this part). Simply continue the previous sentence: The modification process is very simple: by dipping the wood in D4H modification solution it can be obtained highly hydrophobic wood with a water contact angle (WCA) of 140.1° and withstand UV radiation and anti-fouling properties.

Response 6: We deeply appreciate the reviewer's suggestion in such detail. We have corrected it in the new submitted manuscript.

Point 7: Hydrophobic modification of wood surface can be achieved by simply immersing wood in D4H modification solution for 5 minutes. (modification … modification, wood … wood; please avoid repetitions by simply write in D4H solution and immersing samples).

Response 7: Thank you very much for your suggestion. We have corrected it in the new submitted manuscript.

Point 8: Observation of surface micromorphological changes before and after wood modification by scanning electron microscopy (SEM, S4800). (The verb is lacking in this sentence. For example you can use: was performed by scanning electron microscopy).

Response 8: We deeply appreciate the reviewer's suggestion in such detail. We have corrected it in the new submitted manuscript.

Point 9: The analysis of wood was characterised through X-ray spectroscopy (EDS, EX-250).

Analysis was made or was performed not was characterised. EDS is energy dispersive spectroscopy.

Response 9: We deeply appreciate the reviewer's suggestion in such detail. We have corrected it in the new submitted manuscript.

Point 10: In the abstract generally is better to avoid abbreviation (SEM-EDS, XPS, FTIR).

Response 10: Thank you very much for your suggestion. We have corrected it in the new submitted manuscript.

Point 11: This paragraph is very poor: the techniques are not explained, the measurement modalities are not detailed, the experimental conditions are not reported.

Response 11: Thank you very much for your suggestion. We have corrected it in the new submitted manuscript.

Point 12: As shown in movie S1. There is not any movie annexed to the paper. Anyway I think that it is not useful for the comprehension of the research..

Response 12: We deeply appreciate the reviewer's suggestion and comment. In fact, movie files and other data graphs have uploaded to the Polymers journal. In addition, the movie file was provided to confirm the dehydrogenation reaction of wood in D4H solution.

Point 13: Caption of figure 1 should be: Schematic representation of the process for obtaining the D4H modified wood.

Response 13: Thank you very much for your suggestion. We have corrected it in the new submitted manuscript.

Point 14: And the absorption peaks at 1743 cm-1 and 1465 cm-1 were belongs (belong or can be attributed/assigned)! Later in the same paragraph: at 2168 cm-1 is belongs (belongs or is assigned)! At 762 cm-1 is belongs (belongs or is assigned)!.

Response 14: Thank you very much for your suggestion. We have corrected it in the new submitted manuscript.

Point 15: O1s spectra was resolved into spectra of various … Spectra is plural so you have to use were resolved.

Response 15: Thank you very much for your suggestion. We have corrected it in the new submitted manuscript.

Point 16: … showed a new O2 …. Appeared at about 532.46 eV. I suggest to use the present for the verbs because you are showing the present results.

Response 16: Thank you very much for your suggestion. We have corrected it in the new submitted manuscript.

Point 17: In contrast, D4H treated wood samples appeared Si element. It should be: In contrast, D4H wood samples exhibit the presence of Si element.

Response 17: Thank you very much for your suggestion. We have corrected it in the new submitted manuscript.

Point 18: In the inset of figure 4 the chemical elements are reported with the letter K that, I suppose, is the line (Kα). I suggest to remove it or to write in the correct form (for example with a space between the letter of the chemical element and K).

Response 18: Thank you very much for your suggestion. We have corrected it in the new submitted manuscript.

Point 19: The sentence: The WCA could represent the hydrophobicity, it quite obvious and I suggest to remove it.

Response 19: Thank you very much for your suggestion. We have corrected it in the new submitted manuscript.

Point 20:  Did not change much. It is better to write do not change significantly.

Response 20: Thank you very much for your suggestion. We have corrected it in the new submitted manuscript.

Point 21: The durability test of D4H modified wood surface. Firstly, and explanation of the test must be supplied in Materials and Methods. Secondly, I have great doubts about the finger wiping test. It is a completely arbitrary evaluation that cannot be reproduced and repeated because it depends on uncontrolled factors. According to my opinion it is not a scientific test and I don’t know (I made a research on this) standards reporting the finger test. I propose to remove this test.

Response 21: Thank you very much for your suggestion. We have added these details in the new submitted manuscript. I agree with your opinion. However, in fact, in the field of wood hydrophobic modification, there is no standard test method for evaluating the durability of hydrophobic coatings on wood surfaces. In this study, we briefly evaluated the durability of hydrophobic coatings on D4H modified wood surfaces based on the research reports of other scholars (Wu Y , Jia S , Shuang W , et al. A facile and novel emulsion for efficient and convenient fabrication of durable superhydrophobic materials. Chemical Engineering Journal, 2017, 328) and (Lin, W.; Huang, Y.; Li, J.; Liu, Z.; Yang, W.; Li, R.; Chen, H.; Zhang, X. Preparation of highly hydrophobic and anti-fouling wood using poly(methylhydrogen)siloxane. Cellulose 2018, 25, 7341-7353).

Point 22: The tape peeling test, on the other hand, is a standard test that can be used to evaluate the resistance to peeling of a treated surface. Peel tests are used to assess bond quality since the predominant stress, as the name suggests, is peel or tension. But the tape must be weighted before and after the test (the dimension of the tape must be precise) in order to have a scientific datum to affirm that no material has been removed from the surface. Please, repeat this test by weighting the tape and check the international standards for the test.

Response 22: Thank you very much for your suggestion. Similar to the previous answer. I agree with your opinion. However, this study mainly proposes a method for hydrophobic modification of wood, by dipping the wood in D4H modification solution it can be obtained highly hydrophobic wood. Afterwards, we performed a simple evaluation of the durability of the hydrophobic layer based on the literature. Moreover, D4H is mainly grafted on the wood surface in the form of covalent bonds, forming a thin hydrophobic layer. Therefore, we did not conduct a very detailed study on the durability of the hydrophobic film on the wood surface, but only briefly evaluated the durability of the hydrophobic film on the D4H modified wood surface based on the research reports of others.

Point 23: Moreover, the colour change of D4H modified wood was significantly decreased compared to the original wood after UV irradiation. Did the authors make colour measurements before and after ageing? No information about this is reported in Materials and Methods: nor for the UV irradiation nor for the colour measurements. The authors must add such experimental part in Materials and Methods, or they have to remove it from the results.

Moreover, the sentence is completely wrong. I suppose it should be: Moreover, the colour changes due to UV irradiation, in D4H modified wood are much lower that those undergone in unmodified wood.

Response 23: We deeply appreciate the reviewer's suggestion in such detail. We have corrected it in the new submitted manuscript.

Point 24: Water absorption: How this test has been performed? The authors must explain it in Materials and Methods. This test seems to me quite unnecessary.

Response 24: Thank you very much for your suggestion. In fact, the reason why we conducted a 24-h water absorption test on wood before and after D4H modification is to prove that D4H modification can significantly improve the water repellency of wood.

Point 25: Anti-fouling test: also in this case the authors use an empiric test. How did the authors evaluate the absence/presence of contaminants on the surfaces of wood samples? Only by visual observation? This seems to me no scientific and not acceptable in a paper published on Polymer. So, or the authors use scientific methods for evaluated the presence or absence of residues, otherwise this test must be removed.

Response 25: Thank you very much for your suggestion. I agree with your opinion. In fact, we simply analyzed the anti-fouling properties of wood before and after modification from a macro perspective. By taking real photos, we can clearly observe that there is no trace of residual droplets on the surface of  D4H treated wood. This part of the test is also based on the research reports of other scholars. (Lin W , Zhang X , Cai Q , et al. Dehydrogenation-driven assembly of transparent and durable superhydrophobic ORMOSIL coatings on cellulose-based substrates[J]. Cellulose, 2020, 27(1)), (Wu Y , Jia S , Shuang W , et al. A facile and novel emulsion for efficient and convenient fabrication of durable superhydrophobic materials. Chemical Engineering Journal, 2017, 328)

Point 26: In conclusion, the paper is potentially interesting, but it must be deeply revised to have a possibility of being published in Polymers.

Response 26: Thank you very much for your suggestion. we have made a lot of revisions in the new submitted manuscript.

Round 2

Reviewer 1 Report

Hydrophobic modification of wood using tetramethylcyclo-tetrasiloxane

Modification of the wood surface in order to make it hydrophobic is one of the important directions of wood research. These issues have been the subject of many studies. Nevertheless, the research conducted as part of the reviewed paper broadens the scope of knowledge in this topic. In this context, the subject of the article should be considered interesting and up-to-date.

With regard to the previous version of the article, the authors took into account most changes suggested by the reviewer. The article has been changed and supplemented. In my opinion, the article may be published in this version.

Author Response

Point 1: Modification of the wood surface in order to make it hydrophobic is one of the important directions of wood research. These issues have been the subject of many studies. Nevertheless, the research conducted as part of the reviewed paper broadens the scope of knowledge in this topic. In this context, the subject of the article should be considered interesting and up-to-date.

Response 1: We deeply appreciate the reviewer's suggestion and comment.

Point 2: With regard to the previous version of the article, the authors took into account most changes suggested by the reviewer. The article has been changed and supplemented. In my opinion, the article may be published in this version.

Response 2: Thank you very much for your suggestion.

Reviewer 2 Report

In order to improve the integrity and grade of the manuscript, it is suggested to add the experiment of anti-organic solvents (ethanol, acetone, tetrahydrofuran, n-hexane, nitrogen, dimethylformamide, etc.), the change of contact angle under different pH values, and the change of contact Angle under UV lamp at different times to study its anti-UV performance.The experimental method can refer to the literature  of“ Surface physical structure and durability of superhydrophobic wood surface with epoxy resin”.

other recommendation:Please ensure that your response is reflected in the main text and you should use this space to document the changes you make to the original manuscript.  Please highlight new text in the main document using a different coloured font or shading.  

Author Response

Point 1: In order to improve the integrity and grade of the manuscript, it is suggested to add the experiment of anti-organic solvents (ethanol, acetone, tetrahydrofuran, n-hexane, nitrogen, dimethylformamide, etc.), the change of contact angle under different pH values, and the change of contact Angle under UV lamp at different times to study its anti-UV performance. The experimental method can refer to the literature of“ Surface physical structure and durability of superhydrophobic wood surface with epoxy resin”.

Response 1: We deeply appreciate the reviewer's suggestion and comment. Based on your suggestion, the hydrophobized wood prepared in this study has been tested for acid and alkali resistance and resistance to UV radiation. Changes in contact angle of D4H-modified wood after immersion in different pH solutions are provided in the supporting information. UV resistance test was showed in Figure 7 in the manuscript. In addition, we found that the organic solvent resistance (n-hexane, tetrahydrofuran and ethanol) of D4H modified wood was not good, and the hydrophobicity decreased rapidly. In practical applications, wood is mainly affected by some common liquids and acid rain, while D4H modified wood has certain acid resistance and anti-fouling property. Therefore, I think the hydrophobic wood prepared in this study has certain application prospects.

Figure S1 Broken line graph of WCA of the 25% D4H modified wood after having been immersed in different pH solution (pH = 1, 3, 12) with different times.

Point 2: other recommendation:Please ensure that your response is reflected in the main text and you should use this space to document the changes you make to the original manuscript. Please highlight new text in the main document using a different coloured font or shading. 

Response 2: Thank you very much for your suggestion. we have made revisions in the new submitted manuscript, and the revised parts use red fonts.

Reviewer 3 Report

I carefull read the revised version of the paper and the reply to my comments.

I appreciated the authors' reply to the comments and the corrections that they made to the manuscript.

So, I propose to accept the paper. 

Author Response

Point 1: I carefully read the revised version of the paper and the reply to my comments. I appreciated the authors' reply to the comments and the corrections that they made to the manuscript. So, I propose to accept the paper.

Response 1: We deeply appreciate the reviewer's suggestion and comment.

Round 3

Reviewer 2 Report

The article may be published in this version. However, I still suggest that the authors could carry out the study on superhydrophobicity of wood from different perspectives and mechanisms in the future.